# Model of Ploughing Cortical Bone with Single-Point Diamond Tool

**DOI:** 10.3390/ma14216530

**Published:** 2021-10-29

**Authors:** Jing Ni, Yang Wang, Zhen Meng, Jun Cai, Kai Feng, Hongwei Zhang

**Affiliations:** School of Mechanical Engineering, Hangzhou Dianzi University, No. 1158, Jianggan District, Hangzhou 310005, China; nj2000@hdu.edu.cn (J.N.); 202010082@hdu.edu.cn (Y.W.); qq851540319@hdu.edu.cn (J.C.); fengkai@hdu.edu.cn (K.F.); had-zhw@hdu.edu.cn (H.Z.)

**Keywords:** ploughing force model, single-point diamond tool, cortical bone, ploughing coefficient

## Abstract

Generating topological microstructures on the surface of cortical bone to establish a suitable microenvironment can guide bone cells to achieve bone repair. Single-point diamond tools (SPDTs) have advantages in efficiency and flexibility to fabricate surface microstructures. However, the cutting force during ploughing cannot be predicted and controlled due to the special properties of cortical bone. In this paper, a novel cutting model for ploughing cortical bone using an SPDT was established, and we comprehensively considered the shear stress anisotropy of the bone material and the proportional relationship between the normal force and the tangential force. Then, the orthogonal cutting experiment was used to verify the model. The results show that the error of calculated value and the experimental data is less than 5%. The proposed model can be used to assist the fabrication of microstructures on cortical bone surface using an SPDT.

## 1. Introduction

The topological structure of the bone material surface can contribute to the directed differentiation of bone cells into osteoblasts as reported by Langer and Vacanti [1]. Hence, Alison et al. asserted that determining how to construct a suitable topological structure to achieve desired functions has become the latest research goal [2].

The traditional preparation methods mainly included sandblasting, polishing, vapor deposition, and so on [3,4,5]. The single-point diamond tool (SPDT) is more efficient and flexible than the above techniques. Jing Ni et al. fabricated bionic microstructure on the rake face of broach by SPDT to improve processing accuracy and efficiency [6]. There are also many microstructure production research studies involving ploughing microgrooves. Chaochang Chen et al. machined the surface morphology of the elastomer pad using an SPDT and generated a higher quality of pad surface [7]. Quanli Zhang et al. created the indentation method and used SPDT to study the surface damage mechanism of WC/Co materials. The material removal mechanism and surface integrity were evaluated, including plastic deformation, crack formation and propagation, and grain shedding [8]. Jasinevicius discussed the influence of cutting conditions in the machining of semiconductors crystals with a single-point diamond tool based on the quantitative dependence of brittle-to-ductile transition upon the transition pressure value in single-point diamond turning [9]. However, the cutting force by SPDT cannot be predicted and controlled, which affected the further improvement of preparation accuracy of the microstructure.

Otherwise, the mechanical properties of cortical bone completely depend on the directional compact arrangement of the Haversian system in cortical bone, so the mechanical properties of cortical bone show anisotropy [10]. As a kind of typical anisotropic material, the cutting force model of cortical bone has become an attractive research area [11]. Qisen Chen et al. proposed a cortical bone milling force model based on the orthogonal cutting distribution method to improve bone milling operations, which explained the effect of bone material anisotropy on milling force [12]. Based on the orthogonal cutting data, Zhirong Liao et al. combined the furrow effect with the milling force after comprehensively considering the structural characteristics of the cortical bone, the tool geometry, and the ploughing effect and proposed a new bone milling model to calculate the cutting force coefficient [13]. Xiaofan Bai et al. established a cortical bone drilling force model to study the effect of low-frequency axial vibration on bone drilling, drilling force, and temperature rise [14]. When studying the material removal mechanism of bone grinding, Babbar et al. pointed out that the ploughing phase of a single abrasive grain only product lines and did not cause the removal of bone material [15]. Santiuste et al. studied the orthogonal cutting modeling of cortical bone and equated cortical bone as a composite material [16]. The influence of anisotropy on cutting is analyzed and compared with the isotropic method. The results showed that the anisotropy of the cortical bone as the workpiece affects the morphology and temperature of the chip. Hage and Hamade used artificial intelligence to study the complex multiscale structural geometric characteristics of bones from the micro level and established a microscopic cutting force model of bones [17]. Through this micro-feature-based method, the accuracy of force and other related parameters prediction during bone cutting can be improved. However, they provide limited insight on the cutting force of cortical bone by SPDT.

Actually, the cutting process by SPDT is closer to ploughing. Jiwang Yan et al. explored the material removal mechanism of silicon carbide ceramic surface processing using single-point diamond tools [18]. Ayomoh et al. investigated the cutting force at the tip of a single-point diamond tool (SPDT) by predicating on total differentiation of a multivariate function and adaptation of a continuous nonlinear finite series convergent scheme [19]. Heamawatanachai et al. presented an analytical, ductile cutting force model of a novel micromachining tool that was based on micro-orbital motion of a SPDT and verified the model experimentally [20]. In the study of Jumare et al., the influence of various machining parameters on the diamond tool-tip wear during single-point diamond turning (SPDT) of optical grade silicon was examined and proposed a prediction model for single-point diamond tool wear [21]. Yanbin Zhang et al. simplified the grinding process to a single-cone crystal grain and proposed a theoretical force model that considered the mechanism of material removal and plastic accumulation to explore the comprehensive influence of material removal and plastic accumulation on the grinding force model [22]. Min Yang et al. established a ploughing model for the grinding process of ceramic material single diamond particles [23]. Thus, on modeling the ploughing process using an SPDT, the anisotropy and removal mode of cortical bone should be considered.

In this paper, a novel mechanistic model for ploughing force of cortical bone was proposed, which considered the osteon orientation and uncut chip thickness along with the bone material specificity. Then, the orthogonal cutting experiments were employed to evaluate the proposed model in varying osteon cutting angle. Based on the proposed model, the novel preparation of microstructures using an SPDT on the surface of cortical bone can be guided.

## 2. Modeling on Bone Cutting Force for SPDT

The ploughing process of SPDT can be divided into two steps as presented in Figure 1. As shown in Figure 1a, the SPDT is fed along the *Z* axis. During this process, the bone is applied with a force perpendicular to the Haversian system, which is provided by the SPDT. The Haversian lamellae fractures layer by layer when the Haversian system reach the yield limit. As shown in Figure 1b, the SPDT is fed along the *XOY* plane. During this process, the ploughing force is arranged with the Haversian system at the angle *θ*. The cortical bone in the forward direction of the SPDT is subjected to compressive stress and shear stress.

### 2.1. Modeling on Normal Force

When the normal force exceeded a critical value, the cortical bone would irreversibly fracture yield in the contact area during ploughing. The bone tissue was destroyed by the alternating effect of shear stress and yield stress. So, the furrow was carved on the cortical bone surface.

Based on contact mechanics [24], the contact between SPDT and cortical bone is equivalent to that of a rigid cone and elastic space body. The normal force can be expressed as:(1)$$F_{Z}=\frac{2}{\pi }E\frac{h^{2}}{\tan\varphi }$$
where *h* is the uncut chip thickness (UCT) and *E* is the elastic modulus of bovine cortical bone, as listed in Table 1. *φ* is the angle between the cone side and the plane of the cortical bone, as shown in Figure 1;

The relationship between the *h* and the indentation diameter of SPDT *d* can be expressed as:(2)$$h=\frac{d}{2}\tan\varphi$$

The normal force should be expressed as:(3)$$F_{Z}=\frac{\tan\varphi }{2}EA$$

When the *h* exceeds the yield limit, *F_z_* can be calculated as follows:(4)$$F_{Z}=\delta_{s}A=\frac{\pi h^{2}}{8\tan^{2}\varphi }\delta_{s}$$
where *A* is the projected area of the contact surface on the horizontal plane and *δ_s_* is the yield strength of the cortical bone, as shown in Table 1.

The above formula showed that the *F*_z_ is mainly determined by *A* and *h*, so *F_z_* can be expressed as (5):(5)$$F_{z}=\int_{0}^{h}A\delta_{s}dh=\int_{0}^{h}\frac{\pi h^{2}}{8\tan^{2}\varphi }\delta_{s}dh=\frac{\pi }{24\tan^{2}\varphi }\delta_{s}h^{3}=\frac{\delta_{s}V}{8\tan^{2}\varphi }$$

Due to the multilevel micro–nano structure in the cortical bone material, the yield strength *δ_s_* of each depth shows differences, which makes it difficult to accurately express the feed state of the SPDT. Therefore, by further simplifying the above formula, a probability statistical regression analysis model about *F*_z_ and *V* can be obtained. *F_z_* can be fitted as:(6)$$F_{z}=CV^{\alpha }$$
where *C* and *α* are the undetermined coefficients to be fitted.

### 2.2. Modeling on Tangential Force

Considering the anisotropy of cortical bone [27], the shear stress *τ_p_*(*θ*) of the cortical bone orthogonal cutting can be expressed as:(7)$$\tau_{p}(\theta )=(C_{1}+C_{2}\sin(C_{3}\theta +\theta_{0}))\tau_{s}$$
where *τ_s_* is the inherent shear strength of cortical bone, as shown in Table 1. *θ* is the angle between the cutting direction and the Haversian canal. *C*_1_, *C*_2_, and *C*_3_ are fixed coefficients that need to be calibrated.

When the cortical bone undergoes shear fracture, the shear force on the cortical bone can be expressed as:(8)$$F_{p}=A_{p}\tau_{p}=A_{p}(C_{1}+C_{2}\sin(C_{3}\theta +\theta_{0}))\tau_{s}$$
where *A_p_* is the projected area of the tool forward direction, *A_p_* = *h*^2^/tan *φ*. *F_p_* can be calculated as follows:(9)$$F_{p}=\frac{h^{2}}{\tan\varphi }(C_{1}+C_{2}\sin(C_{3}\theta +\theta_{0}))\tau_{s}$$

According to the study of Xipeng Xu et al. [25], the ratio of *F_z_* to *F_p_* is a constant, which can be expressed as ploughing coefficient:(10)$$f_{p}(\theta )=\frac{F_{p}(\theta )}{F_{z}}$$

*F_p_* and *F_z_* have been obtained, so *f_p_*(*θ*) can be calculated as:(11)$$f_{p}(\theta )=\frac{F_{p}(\theta )}{F_{z}}=\frac{h^{2}}{CV^{\alpha }}(C_{1}+C_{2}\sin(C_{3}\theta +\theta_{0}))\tau_{s}$$

*f_p_*(*θ*_0_) can be express as:(12)$$f_{p}(\theta_{0})=\frac{h^{2}\tau_{s}}{CV^{\alpha }}$$

Combining Equation (12) and Equation (11), *f_p_*(*θ*) can be express as:(13)$$f_{p}(\theta )=f_{p}(\theta_{0})(C_{1}+C_{2}\sin(C_{3}\theta +\theta_{0}))$$

Therefore, the tangential force *F_p_*(*θ*) can be expressed as:(14)$$F_{p}(\theta )=f_{p}(\theta )F_{z}=f_{p}(\theta_{0})(C_{1}+C_{2}\sin(C_{3}\theta +\theta_{0}))V^{\alpha }$$

## 3. Experimental Setup and Method

In this paper, the orthogonal cutting trials have been performed using a three-axis miniature machine. The composition of the experimental installation is shown in Figure 2. The SPDT with length L_0_ = 47 mm, shank length L_1_ = 22.5 mm, shank diameter d_1_ = 9.7 mm, tool bit length L_2_ = 22.5 mm, diameter d_2_ = 11 mm, cone angle 90°, and diamond tip diameter L_3_ = 2 mm was employed in an orthogonal cutting experiment. The tip and the bit were welded together by cold welding. The tool specifications could be considered relevant for microstructure surfaces in real surgery application. The cutting force was obtained by a Kistler sensor with a charge amplifier (Type 5080A) and a data acquisition system for force measurement (Type 5697A1) connected to the computer. The experimental data is collected in real time through Dyno-Ware data collection software, under sampling frequency of 1 kHz. As listed in Table 1, the properties of bovine bones are similar to those of human bones. Therefore, adult bovine femurs were selected to prepare the experimental material, and the sample dimension was 100 mm × 20 mm × 5 mm. The prepared cortical bone material is kept in a physiological saline environment to maintain its mechanical properties. After machining experiments, the cut surfaces were analyzed by scanning electron microscopy (COXEM-EM-30-PLUS) to observe the cutting damage on the bone material.

To validate the cutting force model, the ploughing tests for measuring cutting forces have been performed on three-axis miniature machine tool (Figure 2). A series of full factorial of ploughing tests were carried out ploughing length of 10 mm under different uncut chip thickness (*h* = 0.5 mm/0.6 mm/0.7 mm/0.8 mm) and cutting angles (*θ* = 0°/45°/90° Figure 3). The ploughing speed was fixed at 100 mm/min to reduce the influence caused by the thermal effect of bones during ploughing. The details of cutting parameters can be seen in Table 2, where tests were repeated five times for ensuring data stability. Since the long axes of femur and its osteons are in the same direction, here, cutting angle could be referred to the angle between osteon direction of the bone long axes and feed direction of SPDT. *F*-test and residual analysis were used to analyze the significance of the collected experimental data and predicted data [28]. Hampel filtering was used by MATLAB to filter out the abnormal values in the experimental data to solve the data abnormal problem caused by the accuracy error of the experimental equipment, and the filtering spacing was 5 [29].

## 4. Results and Discussion

### 4.1. Morphology of Bone Sample

The SEM images of the ploughing morphologies of the SPDT at various angles are shown in Figure 3. One can see in the morphology of indentation of the SPDT, shown in Figure 4, that the surface of the cortical bone has an irreversible plastic deformation indentation. As shown in Figure 3, for the 0° angle, the direction of the cracks was basically the same as that of the Haversian canal. For the 45° angle, the cracks were obviously jagged. For the 90° angle, the angle between the crack and the Haversian canal increased further. The direction of bone cracks and Haversian canal showed great differences at different cutting angles.

### 4.2. Normal Force Validation

To calibrate the proposed bone single-point normal force model (Equation (6)), the indentation tests were employed. Figure 5a is regression fitting curve between the volume pressed in cortical bone and the normal force. The relationship between the volume *V* of SPDT pressed into the cortical bone and UTC *h* is as follows: $V=\pi h^{3}/3$. For each volume of 0.05 mm^3^, a segment of experimental data was selected and represented by symbol ‘☆’. It can be seen from Figure 5a that the normal force (*F_z_*) and volume (*V*) have a good exponential distribution, and the increasing trend of *F_z_* gradually slows down.

With the cutting force data under various UCT obtained from orthogonal cutting experiments, the proposed model (Equation (6)) was calibrated (Table 3) with the weighted least squares optimization algorithm where the *R*^2^ was 0.99977, indicating the reasonable adequacy of the proposed model where the coefficients *C* and *α* were calculated as 707.53 and 0.39, respectively. The error value of the regression model was analyzed. It can be seen from Figure 5b that, under various volumes, the corresponding error rate is always controlled below 5%. It means that there is no force mutation on the surface of cortical bone, and the bone has undergone elastoplastic deformation.

The comparison between experimental value and predicted value of volume corresponding to UCT (0.5 mm/0.6 mm/0.8 mm) is shown in Figure 6a. Figure 6b shows the relative error rate at the three volumes where the blue points represent the error rate of the corresponding volume. It can be clearly seen that under the experiment of different uncut thickness, the calculated value and the experimental value are basically consistent and in line with the expected trend of the model. The error fluctuates greatly at first and then gradually stabilizes; the overall error rate of the calculated value is low (less than 5%), which is within the acceptable error range. The proposed model can truly represent the true state of the single-point diamond when the cortical bone was pressed. It can be considered that at the significance level of 0.05, the empirical model of normal force is *F_z_* = 707.53 *V*
^0.39^.

As shown in Figure 7a, the predicted values (*F_z_*) of the traditional Johnson–Cook model and proposed model was compared. Obviously, the proposed model has better consistency with the experimental results, while the Johnson–Cook model is significantly different from the experimental results. This is due to the fact that the Johnson–Cook model only considers the mechanical properties of homogeneous materials but has a disadvantage in specification for mechanical properties of complex anisotropic materials.

### 4.3. Tangential Force Validation

To calibrate the proposed tangential force model (Equation (11)), the orthogonal cutting tests were employed by changing the UCT from 0 mm to 0.7 mm and the cutting angle from 0° to 90°, fitting the tangential force with a regression algorithm. It can be seen from Figure 8 that when the UCT is 0.7 mm, *F_z_* can be regarded as basically the same under different cutting angles (*θ* = 0°/45°/90°), but *F_p_* shows great anisotropy and *F_p_* and *F_z_* show an obvious proportional relationship. Ideally, the ploughing coefficient (*f_p_* (*θ*)) of cortical bone should be an invariant constant under various UCT. However, large anisotropy occurs at different cutting angles, and this constant is only related to the performance parameters of cortical bone itself. This anisotropy is generated by the arrangement of the orientation Haversian system cortical bone. The influence of this anisotropy on the ploughing of cortical bone cannot be ignored. When the angle is 0°, the ploughing coefficient is the smallest (*f_p_* (0°) = 0.29); as the angle increases, the ploughing coefficient gradually increases, reaching the maximum value at 45° (*f_p_* (45°) = 1.21). Then, it decreases as the angle continues to increase (*f_p_* (90°) = 0.55). With the tangential force data achieved under various osteon cutting angle and UCT obtained from orthogonal cutting experiments, the proposed ploughing coefficient (Equation (11)) and tangential force model (Equation (12)) were calibrated (Table 4) with the least squares optimization algorithm where the *R*^2^ is 0.95, indicating the reasonable adequacy of the proposed model.

Table 5 shows the relative error rate of tangential force in different cutting angles under UCT of 0.7 mm. As listed in Table 5, the ploughing coefficient *f_p_*(*θ*) shows obvious anisotropy at different angles (*f_p_*(90°) ≈ 2 *f_p_*(0°), *f_p_*(45°) ≈ 4 *f_p_*(0°)). Under the same cutting angles and different UCT, the ploughing coefficient was basically the same, and the error rate was low (the highest was 2.49%, the lowest was 1.00%). *f_p_*(θ) can be expressed as: *f_p_* (*θ*) = 0.29(0.294 + 0.934 sin (0.45*θ*)). *F_p_*(*θ*) can be expressed as *F_p_*(*θ*) = *f_p_* (*θ*)*F_z_* = 208(0.294 + 0.934 sin(0.45*θ*)) *V*
^0.39^.

The tangential force was calculated using this model, and the calculated value was compared with the experimental value. Figure 9 shows the residual distribution between the calculated value and measured values of tangential force under various cutting angles. The residual value was represented by a blue triangle. As shown in Figure 9, the residual value was small. Most of the residual values were distributed in (−3,3) within the interval when the cutting angle was 0°. With the cutting angle increased (*θ* = 45°/90°), the residual value increased, and the residual values were distributed in (−4,4) within the interval. The residual value fluctuated in a narrow range, and the calculated value basically coincides with the experimental value.

Table 6 is the error rate of the predicted value under different angles. As shown in Table 6, the value of *F_p_*(θ) varies greatly under different angles by the anisotropy of cortical bone. The relative error rate of the tangential force was 4.36% at the maximum and 0.05% at the minimum under different cutting angles. The error fluctuation is within the allowable range. The relative error rate of the analysis results was below 5%. It can be considered at the significance level of 0.05 that the tangential force model in cortical bone ploughing achieves the expected results. The model can better reflect the true state of the tangential force of cortical bone at different angles.

The simulation and experimental results showed that the cortical bone tangential force model proposed in this paper was in good agreement with the experimental data at various UCT and cutting angles. The model can accurately represent the influence of cortical bone anisotropy on the ploughing coefficient in the ploughing process.

From Figure 7b–d, it can be seen that the predicted results of the tangential forces for the ploughing (*F_p_*) from proposed model yield a closer value to the experimental results compared with those of the traditional isotropy model (Johnson–Cook model). The predicted results of the proposed model are also in conformity with the experimental results in the changing trend of cutting force. Moreover, a significant difference of cutting forces can be observed in different osteon cutting angles experimentally; the proposed model captures this well with respect to osteon anisotropy, whereas the Johnson–Cook model only shows only a single estimate value under initial osteon cutting angles. To reduce the influence caused by the thermal effect of bones during ploughing, the ploughing speed was fixed at low speed. The two factors of cutting speed and cutting zone temperature were not studied. In further research, the effect of high-speed ploughing on cutting force and temperature will be considered to complement the research in this field.

## 5. Conclusions

In this paper, a novel mechanistic model was developed for predicting the cutting forces in the bone ploughing process. First, the normal force and tangential force in cortical bone ploughing were modeled by considering the mechanical properties of cortical bone and its anisotropic multilevel micro–nano structure. Second, a single-point diamond tool orthogonal cutting experiment was designed to verify and revise the model. Third, the accuracy of the model was analyzed using statistical analysis methods. The prediction results of the model agree well with the experimental results, indicating that it has the potential to make the topological microstructures on the surface of cortical bone, optimize cutting parameters, and guide the design of orthopedic tools. The main findings of the paper can be summarized as follows:

1. The normal force of ploughing process can be regarded as related to the volume of the SPDT pressed into the cortical bone, and the relationship between the normal force and volume can be obtained by *F_z_* = 707.53 *V*
^0.39^. The tangential force *F_p_* and the normal force *F_z_* have a fixed proportional relationship, which can be expressed by *F_p_* (*θ*)= *f_p_* (*θ*) *F_z_*= 208 (0.294 + 0.934 sin(0.45*θ*)) *V*
^0.39^.

2. Orthogonal cutting experiment results show that material performance showed greater anisotropy under different cutting angles. This anisotropy is manifested in the difference in *f_p_*(*θ*). The *f*_p_(*θ*) values under different cutting angles have the relationship as follow: *f_p_*(90°) ≈ 2 *f_p_*(0°), *f_p_*(45°) ≈ 4 *f_p_*(0°). It can be found that tangential force has a higher *f_p_* (*θ*) at 45° osteon cutting angle compared with 0° and 90° osteon cutting angles.

3. The prediction error lower than 5% on normal force and a maximum prediction error of 17.79% and minimum error of 0.05% on tangential force of ploughing were reached using the proposed model.

4. The proposed model provides a new theoretical solution for the preparation of topological microstructures on the surface of cortical bone in modern medical treatment to accelerate bone repair. It has laid a certain theoretical foundation for the optimization of the processing parameters of biobone materials in modern medicine.

With proposed model, the researchers could simulate the cutting force under different cutting conditions (e.g., cutting angles and uncut chip thickness) and directions (e.g., parallel, inclined, or perpendicular to the bone long axis) before the microstructure is constructed to select the optimized cutting parameters to allow reduction of the cutting force and reduction of bone damage.

## Figures and Tables

**Figure 1 materials-14-06530-f001:**
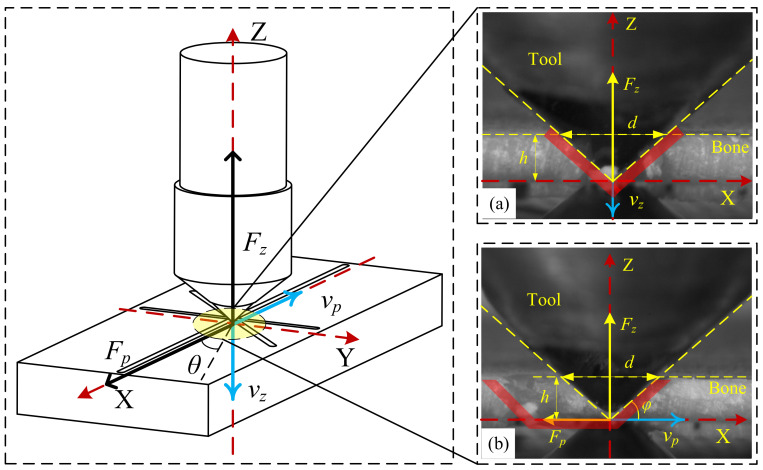
(**a**,**b**) Ploughing principle diagram.

**Figure 2 materials-14-06530-f002:**
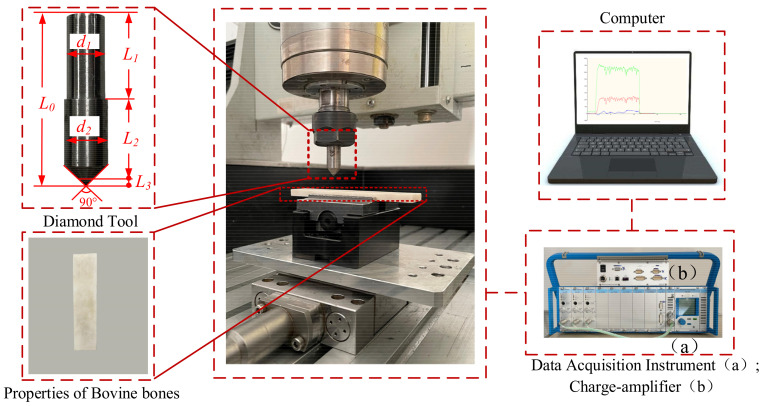
Experimental setup.

**Figure 3 materials-14-06530-f003:**
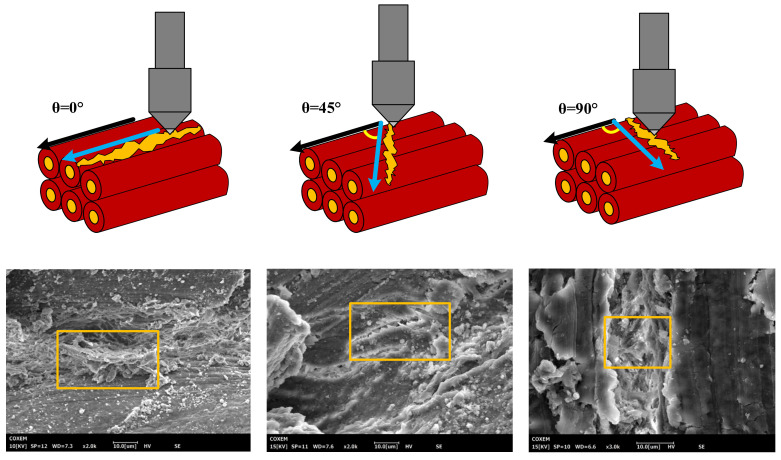
Cutting angle between tool and Haversian canal and its morphology of furrow.

**Figure 4 materials-14-06530-f004:**
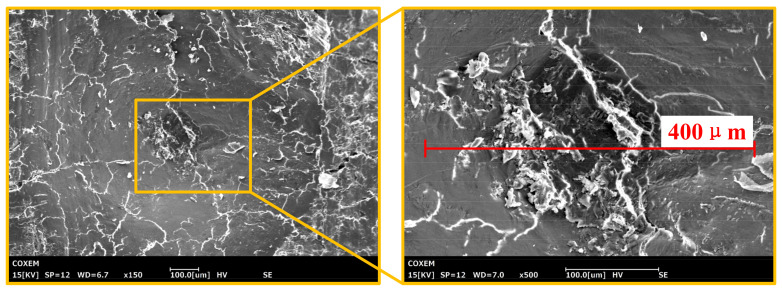
Morphology of indentation of the SPDT.

**Figure 5 materials-14-06530-f005:**
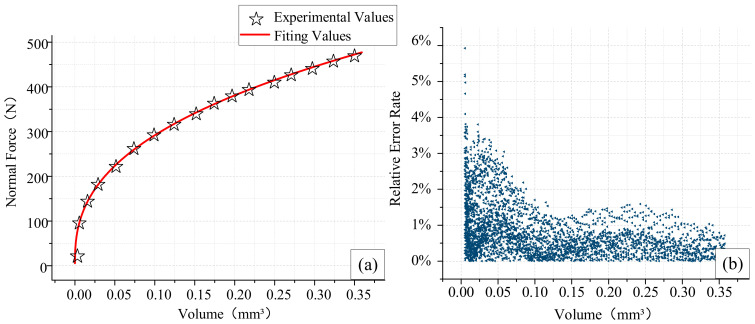
Regression fitting curve between the volume pressed in cortical bone and the normal force (**a**); the relative error rate of the fitting model (**b**).

**Figure 6 materials-14-06530-f006:**
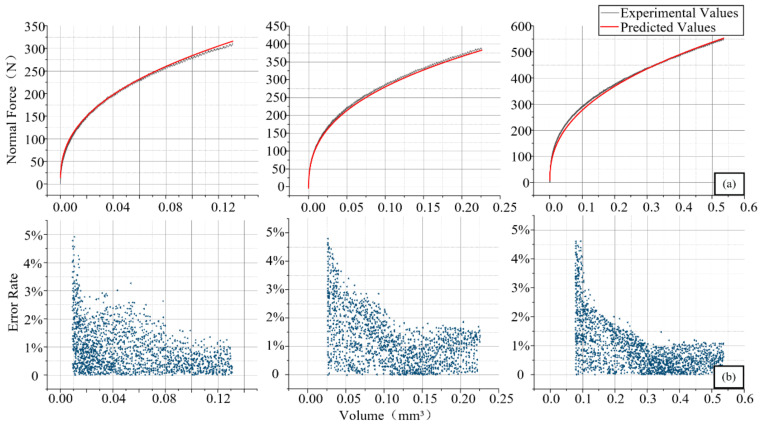
Comparison between experimental value and predicted value of various volumes (**a**); error rate distribution of residual of various volumes (**b**).

**Figure 7 materials-14-06530-f007:**
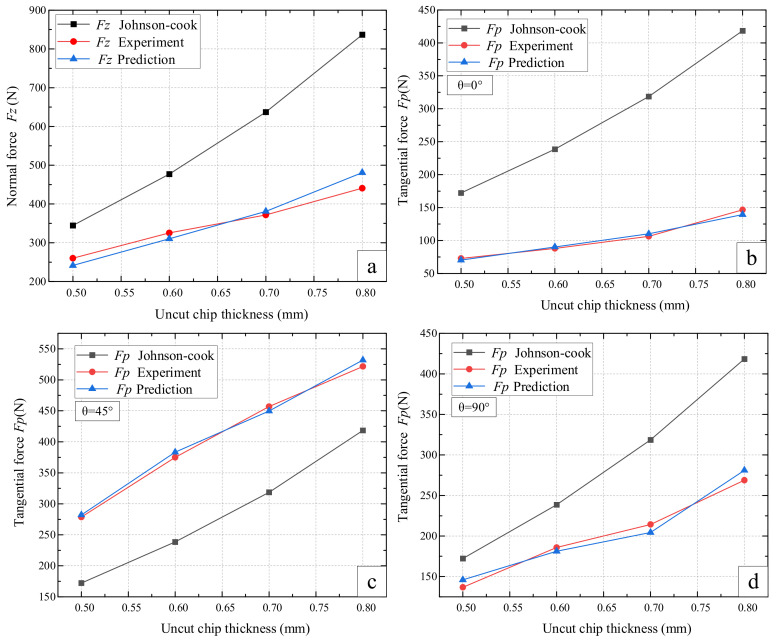
Comparison of isotropy and anisotropy modeling on bone cutting force: (**a**) comparison of different model on normal force; (**b**–**d**) comparison of different modeling on tangential force.

**Figure 8 materials-14-06530-f008:**
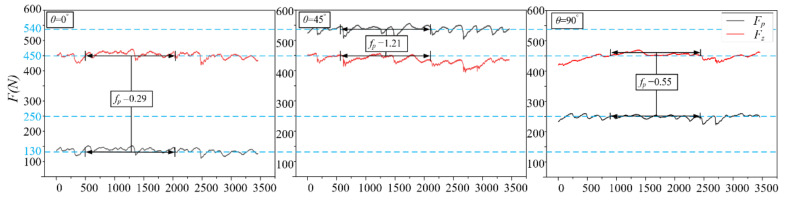
Trend and ratio chart of normal force and tangential force under different cutting angles.

**Figure 9 materials-14-06530-f009:**
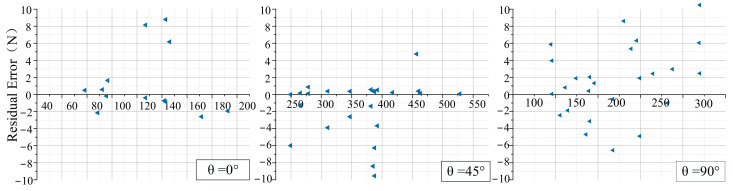
Residual distribution.

**Table 1 materials-14-06530-t001:** Performance parameters of bovine bone and human bone [25,26].

Performance Parameter	Bone Type	
	Bovine	Human
Tensile strength (MPa)	140–250	130–200
Compressive strength (MPa)	45–150	40–145
Young’s modulus (GPa)	10–22	10–17
Shear modulus (MPa)	3	3
Density (kg/m^3^)	1950–2100	1800–2000
Poisson’s ratio	0.33	0.4
Specific heat (J/kg K)	1300	1330
Thermal conductivity (W/m K)	0.1–0.3	0.1–0.43

**Table 2 materials-14-06530-t002:** Experiment parameter.

NO	Uncut Chip Thickness (mm)	Cutting Angle
1	0.5	0°/45°/90°
2	0.6	0°/45°/90°
3	0.7	0°/45°/90°
4	0.8	0°/45°/90°

**Table 3 materials-14-06530-t003:** Significance analysis.

Level: 0.7 mm Fitting: *F_z_ =* 707.53 *V* ^0.39^ *R*-Squared *R*^2^: 0.99977					
	DOF	Sum of Square	Mean-Square	*F*-Measure	Confidence Level
Regression Coefficient	2	2.89 × 10^8^	1.45 × 10^8^	1.44 × 10^−7^	95%
Residual Error	4111	41,283.57	10.04		

**Table 4 materials-14-06530-t004:** Coefficients of proposed bone ploughing coefficient.

Coefficient	*f_p_*(*θ*_0_)	*C* _1_	*C* _2_	*C* _3_	*θ* _0_
Optimized value	0.29	0.294	0.934	0.45	0

**Table 5 materials-14-06530-t005:** The relative error ratio of the three levels.

*f_p_* (0°)	Error Rate	*f_p_* (90°)	Error Rate	*f_p_* (45°)	Error Rate
0.29	1.36%	0.55	1.99%	1.23	2.49%
0.29	1.42%	0.55	1.02%	1.21	1.06%
0.29	2.26%	0.55	1.00%	1.21	1.05%
0.30	1.00%	0.58	1.07%	1.22	1.46%
0.30	1.95%	0.58	1.06%	1.22	1.83%

**Table 6 materials-14-06530-t006:** The error rate of the predicted value under different angles.

**0.5 mm**		**0.6 mm**		**0.8 mm**	
***F_p_* (0°) (N)**	**Error Rate**	***F_p_* (0°) (N)**	**Error Rate**	***F_p_* (0°)**	**Error Rate**
64.35	6.23%	88.61	2.43%	126.84	0.31%
80.04	3.02%	95.58	0.19%	146.06	4.22%
60.56	5.29%	92.24	0.62%	142.96	0.61%
71.70	3.15%	96.63	1.69%	141.90	0.05%
75.46	3.84%	78.30	0.64%	171.60	1.51%
**0.5 mm**		**0.6 mm**		**0.8 mm**	
***F_p_* (45°)**	**Error Rate**	***F_p_* (45°)**	**Error Rate**	***F_p_* (45°)**	**Error Rate**
314.57	1.25%	374.95	1.61%	528.01	1.58%
277.75	0.31%	383.11	0.36%	528.79	0.68%
266.73	0.48%	393.6	2.18%	542.52	0.50%
296.38	1.18%	349.87	0.75%	534.53	0.23%
255.31	0.1%	395.99	0.94%	525.84	1.94%
**0.5 mm**		**0.6 mm**		**0.8 mm**	
***F_p_* (90°)**	**Error Rate**	***F_p_* (90°)**	**Error Rate**	***F_p_* (90°)**	**Error Rate**
145.83	0.04%	172.33	1.09%	282.28	0.36%
136.02	11.06%	194.27	0.68%	299.33	3.75%
164.39	1.13%	188.69	0.22%	266.69	4.36%
155.29	1.57%	190.56	2.52%	244.23	4.54%
128.36	17.79%	160.52	0.50%	312.95	1.89%

## Data Availability

The data presented in this study are available in Appendix A.

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
