# Peer review of "Model of Ploughing Cortical Bone with Single-Point Diamond Tool"

_materials, 2021, doi:10.3390/ma14216530_

Round 1
Reviewer 1 Report
This is an interesting article in which the authors propose a mechanistic model for ploughing force of cortical bone. The authors used adult bovine femur (since performance parameters were similar to human bone) to perform experimental evaluation of the model by orthogonal cutting experiments varying the angle. With this model, it could be possible to stimulate the cutting force under different cutting conditions and directions, to be able to select optimum cutting parameters to reduce cutting force and bone damage. Overall the article is well written and well explained; it presents an optimization of cutting parameters with single-point diamond tool for improving cortical bone repair with minimum damage. I just detected some writing mistakes that could be solved by a careful English edition.
Author Response
Dear reviewer:We sincerely thank the editor and all reviewers for your valuable feedback that we have used to improve the quality of our manuscript entitled “Model of ploughing cortical bone with single point diamond tool”. Thank you for giving us the opportunity to modify our manuscript. This is a great support for our nearly one year’s research work.We have studied reviewers' comments carefully and tried our best to revise our manuscript according to the comments.
Reviewer 2 Report
Here are my comments for the authors:
1. It will be better if the authors perform the designed study at different ploughing speeds to examine the effects of ploughing speeds on the cortical bone study with SPDT.
2. Why have the authors selected 0/45/90 degrees for their study?
3. Why is the error rate high in the 0.5 mm case at 0 and 90 degrees in Table 6?
Reviewer 3 Report
1. This manuscript has a weird structure where the the introduction together with part 2 are extremely extensive, presenting theory on SPDT modelling for no reason. Part 2 is irrelevant to the work. Please create a proper materials and methods section.
2. the experimental setup and method are lacking detail. how were the experiments designed?
3. I guess that you applied some kind of Design of Experiments (DOE) systematic factor analysis. How did you do the statistics?
4.Please detail the repeats and the parameters in each repeat. For example uncut tip thickness is first described on table 2. Readers are surprised to see information in tables that do not exist in the text
5. The authors present a series of irrelevant information, explaining even the F test. The paper needs to be concise without presenting all kinds of background information that could be included in a book.
6. What do the *** significance represent? One more type of information presented in a table without any previous explanation.
7. There is NO discussion about your results. The section is called "results and discussion" but there is nowhere to find the discussion about the significance of your work, the comparison of your results to other published papers and the limitations of your experiments.
Reviewer 4 Report
'Model of ploughing cortical bone with single point diamond tool' is a well structured paper on the use of single pointed diamond tool to create microstructures on bone. The study is well designed and carried out using state of the art techniques. data analysis is appropriate and the results appear sound and convincing. I think the authors should try and make an extra effort to guide readers through the figures by expanding their very coincise legends to figures. Though I understand that figures are actually explained inside the main text, it is helpful for the readers to find some basic info and explanation right in the legend
Round 2
Reviewer 3 Report
The authors have provided sufficient responses to the comments of the previous review round.